# Success of Checkpoint Blockade Paves the Way for Novel Immune Therapy in Malignant Pleural Mesothelioma

**DOI:** 10.3390/cancers15112940

**Published:** 2023-05-27

**Authors:** Lizbeth Rondon, Roberto Fu, Manish R. Patel

**Affiliations:** 1Department of Medicine, Hennepin County Medical Center, Minneapolis, MN 55404, USA; lizbeth.rondonrueda@hcmed.org (L.R.); roberto.fu@hcmed.org (R.F.); 2Division of Hematology, Oncology, and Transplantation, Department of Medicine, University of Minnesota, Minneapolis, MN 55455, USA

**Keywords:** malignant pleural mesothelioma, immune checkpoint inhibitors, PD-1, PD-L1, CTLA-4, B7-H3, mesothelin therapy, CAR-T cells, NK cells

## Abstract

**Simple Summary:**

This review article will focus on the landmark studies that came to define current treatment paradigms for malignant pleural mesothelioma (MPM), all the way from standard chemotherapy to ongoing clinical trials with immunotherapy combinations, along with some of the most novel approaches currently under investigation. Malignant pleural mesothelioma remains an orphan disease, but current and future scientific advancements hold promise for an increased chance of survival and continued quality of life for the thousands of patients diagnosed yearly with this illness. Most recently, in 2022, immune checkpoint blockade has been FDA-approved as first-line therapy for advanced malignant pleural mesothelioma (MPM).

**Abstract:**

Malignant pleural mesothelioma (MPM) is a malignancy associated with asbestos exposure and is typically categorized as an orphan disease. Recent developments in immunotherapy with anti-PD-1 and anti-CTLA-4 antibodies, specifically with agents nivolumab and ipilimumab, have demonstrated an improvement in overall survival over the previous standard chemotherapy leading to their FDA-approval as first-line therapy for unresectable disease. For quite some time, it has been known that these proteins are not the only ones that function as immune checkpoints in human biology, and the hypothesis that MPM is an immunogenic disease has led to an expanding number of studies investigating alternative checkpoint inhibitors and novel immunotherapy for this malignancy. Early trials are also supporting the notion that therapies that target biological molecules on T cells, cancer cells, or that trigger the antitumor activity of other immune cells may represent the future of MPM treatment. Moreover, mesothelin-targeted therapies are thriving in the field, with forthcoming results from multiple trials signaling an improvement in overall survival when combined with other immunotherapy agents. The following manuscript will review the current state of immune therapy for MPM, explore the knowledge gaps in the field, and discuss ongoing novel immunotherapeutic research in early clinical trials.

## 1. Introduction

Malignant pleural mesothelioma (MPM) is a rare disease that stems from the mesothelial cells of the pleura. Three types of MPM are recognized according to their histopathologic findings: epithelioid, biphasic, and sarcomatoid. Its rarity, overall poor prognosis (8–15 months), and limited treatment options have given MPM its status as an orphan disease [1]. It is strongly linked to asbestos exposure, with more than 400 asbestiform fibers in nature, many of which have been proven to be carcinogenic [2]. While the use of asbestos in the industrial world is declining, many people continue to be exposed to it, particularly those living in developing economies and/or working within the construction field, with limited mining regulations and unexploited natural deposits [3]. Due to its long latency period, mesothelioma can take anywhere from 20 to 40 years to manifest symptoms, and its mortality has been estimated at 43,000 deaths worldwide annually, with 2500–3000 new cases occurring each year in the United States [2,4]. 

There are currently no curative approaches for advanced MPM, and treatment is offered with the goal of improving survival and decreasing disease burden. Even for a select few eligible patients, extra-pleural pneumonectomy was not curative for most and is a highly morbid procedure. Since 2002, the standard of therapy has been a combination of cisplatin and pemetrexed based on a randomized trial demonstrating a survival benefit compared to cisplatin alone [5]. Though the response rate is close to 40%, this is almost exclusively in the epithelioid subset, while biphasic and sarcomatoid patients fared much worse [6]. 

Despite many efforts, targeted therapy to cell cycle regulators, growth factors, or angiogenesis pathways has been otherwise ineffective. Even though these pathways have been found to be implicated in the development of MPM in preclinical settings, several compounds targeting these pathways lacked activity in the clinical setting [7]. Nonetheless, results from other solid tumors treated with immune therapy led to an interest in pursuing this approach for MPM. As it is related to asbestos exposure, MPM pathogenesis has always been felt to be inflammatory in nature. This theory, combined with the hypothesis that MPM has a rich T cell microenvironment, suggested that immunotherapy might be effective for this disease [8]. Though the results of clinical trials with immunotherapy have been quite variable, early-phase studies demonstrated some promise for immune checkpoint blockade. This culminated in the pivotal Checkmate 743 trial, in which ipilimumab/nivolumab combination therapy resulted in a survival benefit compared to standard chemotherapy, leading to FDA approval of this regimen for previously untreated, unresectable MPM. In the following manuscript, we will review the data from the most representative published clinical trials of immune checkpoint inhibitors (ICI). We will explore the knowledge deficits in the field, and finally, we will review novel immunotherapy options that are currently undergoing clinical investigation for MPM.

## 2. Early Immunotherapy Trials for MPM

### 2.1. Interferon-Based Therapy

Researchers have been interested in immune therapy for MPM for several decades. The first trials were with interferon (IFN)-based therapy, given that preclinical work demonstrated an immunologic effect of exogenous Type I IFNs in murine models. These trials involved using IFN-α in combination with chemotherapy. While there was clinical activity in these small, single-arm phase II studies, the overall limitation was the toxicity of systemic IFNα therapy. Sterman et al. developed an adenoviral “immune gene” therapy approach that used adenoviral vectors to express IFNα or IFNβ and was administered intra-pleurally [9]. Similarly, tumor regression was observed when IL-2 was administered intra-pleurally, but this was only investigated in a limited number of preliminary trials, and the results were not pursued further [10]. With the discovery of immune checkpoint proteins (e.g., CTLA-4 and PD-1) as key factors in mediating immunologic escape for many cancers, there was renewed interest in studying immune checkpoint blockade in MPM.

### 2.2. Checkpoint Inhibitors

The most widely studied immune checkpoint inhibitors are cytotoxic T-lymphocyte-associated protein 4 (CTLA-4), programmed death protein-1 (PD-1), and programmed death ligand-1 (PD-L1) (Figure 1) The early phase studies using CTLA-4 antibodies in MPM were quite promising, with reports of individual patients having very striking responses [11]. Subsequently, the DETERMINE trial of tremelimumab vs. placebo was started. Unfortunately, this trial failed to prolong overall survival (OS) compared to placebo 7.7 months vs. 7.3 months, hazard ratio (HR) 0.92, [95% confidence interval (CI) 0.76–1.12)] [12]. These findings rendered an end to the continued advancement of anti-CTLA-4 monotherapy in the context of relapse. Importantly, most of the participants in this study were patients with epithelioid histology. 

NIVOMES, CONFIRM, and MERIT trials (Table 1), all investigated Nivolumab, an anti-PD-1 antibody. Out of these, CONFIRM was the largest study as it evaluated 332 patients, the majority of which were treated in the third-line setting. The results showed that the median overall survival was 10.2 months with nivolumab and 6.9 months with placebo (adjusted HR: 0.69 [95% CI: 0.52–0.91]; *p* = 0.0090). The median progression-free survival (PFS) with nivolumab was 3.0 months versus 1.8 months with placebo (adjusted hazard ratio 0.67 [95% CI 0.53–0.85]; *p* = 0.0012). In total, 88% of patients enrolled in this study had epithelioid histology. Though there was no statistical benefit in non-epithelioid histology, the hazard ratio of 0.69 favored nivolumab. Unfortunately, the number of patients with non-epithelioid histology was too small to draw conclusions here. Additionally, PD-L1 expression had no bearing on OS. Infusion-related reactions (six [3%] vs. none) and diarrhea (six [3%] vs. two [2%]) of 221 participants in the nivolumab group were the most frequently reported grade 3 or worse treatment-related side events [13,14,15]. One report of single-institution real-world data of 29 patients showed a 19.5% response rate with immune checkpoint inhibitors. The median PFS and OS in this cohort were 5.5 months and 19.1 months, respectively [16].

With these positive results came the MAPS 2 study, a multicenter, randomized phase II trial evaluating nivolumab versus a combination of nivolumab plus ipilimumab (an anti-CTLA-4 antibody) until disease progression or unacceptable toxicity. This study met its primary endpoint; the 12-week disease control rate (DCR) was 40% in the nivolumab group versus 52% in the combination group. Objective responses were seen in 19% and 28% for nivolumab monotherapy and the combination, respectively. Median OS was 11.9 months (95% CI 6.7–17.7) in the nivolumab group and 15.9 months in the nivolumab plus ipilimumab group. Grade 3–4 toxicity events were seen in 14% in single therapy vs. 26% with combination. Similar to MAPS2, the INITIATE phase II study also evaluated the efficacy of nivolumab with ipilimumab in 34 MPM patients, reporting an objective response rate (ORR) of 29% and DCR of 68% (95% CI 50–83) at 12 weeks [17]. These last findings were not only encouraging but demonstrated ipilimumab’s efficacy when used in combined strategies, leading to new trials that placed the combined immunotherapy agents on the front lines of MPM treatment. 

**Table 1 cancers-15-02940-t001:** Summary of early trials for checkpoint inhibitors.

Trial	Agent	Phase	Target	Patients	Outcome
DETERMINE	Tremelimumab	II	CTLA-4	571	Median OS ^1^ 7.7 months
NIVOMES	Nivolumab	II	PD-1	38	DCR ^2^ 50%
MERIT	Nivolumab	II	PD-1	34	Median OS ^1^ 17.3 and PFS ^3^ 6.1 months
CONFIRM	Nivolumab	III	PD-1	332	Median OS ^1^ 10.1 and PFS ^3^ 3.0 months
KEYNOTE-028	Pembrolizumab	IB	PD-1	25	ORR ^4^ 20%.
Chicago group: NCT02399371	Pembrolizumab	II	PD-1	65	Median OS ^1^ 11.5 and median PFS ^3^ 4.5 months
PROMISE-MESO	Pembrolizumab	III	PD-1	144	OS ^1^ 10.7, PFS ^3^ 2.4 months. ORR ^4^ 22%
JAVELIN	Avelumab	IB	PDL-1	53	ORR ^4^ 9%
DREAM	Durvalumab	II	PDL-1	55	ORR ^4^ 61% and PFS ^3^ 71%
MAPS 2	Nivolumab/ ipilimumab	II	PD-1/ CTLA-4	125	DCR ^2^ 52% and ORR ^4^ 28%. OS ^1^ 15.9 months
INITIATE	Nivolumab/ ipilimumab	II	PD-1/ CTLA-4	38	DCR ^2^ 68% and ORR ^4^ 29%
NIBIT-MESO-1	Tremelimumab/ Durvalumab	II	CTLA-4/ PDL-1	40	Median OS ^1^ 16.6 and median PFS ^3^ 5.7 months

^1.^ Overall survival. ^2.^ Disease control rate. ^3.^ Period-free survival. ^4.^ Objective response rate. References: DETERMINE [12], NIVOMES [15], MERIT [14], KEYNOTE-028 [18], Chicago group [19], PROMISE-MESO [20], JAVELIN [21], DREAM [22], MAPS2 [23], INITIATE [24], NIBIT-MESO1 [25].

Pembrolizumab, another anti-PD-1 antibody, was studied in the KEYNOTE-028 phase 1b trial and subsequently in a phase II trial by the Chicago group. This last trial enrolled 65 previously treated participants in phase II with a median PFS of 4.1 and an OS of 11.5 months. Unfortunately, the phase III randomized study PROMISE-MESO did not meet its primary endpoint. It enrolled 144 PDL-1 unselected patients who had progressed on standard chemotherapy and evaluated pembrolizumab’s efficacy as a second-line agent. Pembrolizumab did not result in longer PFS (2.5 vs. 3.4 months) or longer OS (10.7 vs. 11.7 months) when compared to chemotherapy [17]. This was a well-conducted randomized controlled trial, and the lack of effect was unlikely due to imbalances in the arms. Interestingly, the ORR (22%) and OS in this trial are very similar to that noted in the MAPS2 trial with nivolumab monotherapy. 

## 3. Immunotherapy in the Frontline 

CHECKMATE 743 (CM743) was a multicenter randomized, open-label phase 3 trial of ipilimumab/nivolumab vs. chemotherapy [26]. Participants were randomized to receive nivolumab plus ipilimumab as single therapy vs. platinum plus pemetrexed. In total, 713 patients were enrolled between 2016 and 2018, and 605 of them received a random assignment. This was a positive study with nivolumab plus ipilimumab significantly extending overall survival (median OS 18.1 months [95% CI 16.8–21.4] versus 14.1 months [12.4–16.2]; hazard ratio 0.74 [96.6% CI 0.60–0.91]; *p* = 0·0020). The 3 year OS rate for the nivolumab + ipilimumab group was 23% (95% CI 351–465), and for the chemotherapy group was 15% [27]. Most subgroups benefited from Ipilimumab/Nivolumab treatment. 

Notably, OS was not significantly improved in patients with epithelioid histology or with PDL-1 expression < 1%. A 4-gene inflammatory signature was associated with improved outcomes, as was the lung immune prognostic index (LIPI). There was no unexpected toxicity, and the Grade 3–4 treatment-related adverse events (TRAE) were similar between the groups. In those receiving ipilimumab/nivolumab, the development of TRAE was associated with a better OS, which is consistent with observations with immunotherapy treatment in patients with melanoma and NSCLC [17,28,29]. 

This regimen was finally approved by the FDA in 2020 based on these results. It remains unclear how much benefit there is in patients with epithelioid histology over chemotherapy; however, the ORR, PFS, and OS were similar between chemotherapy and immune therapy, suggesting that this remains a good option in the first-line treatment with epithelioid histology, particularly in patients who may not be suitable candidates for systemic chemotherapy. 

CM743 is now the standard of care; according to the National Comprehensive Cancer Network (NCCN) guidelines, no other immunotherapy alternatives have been approved for MPM treatment. However, there are several other trials evaluating immune checkpoint therapy in the first line for mesothelioma. The Phase II DREAM trial studied the combination of durvalumab (anti-PD-L1) and first-line chemotherapy (cisplatin and pemetrexed). In total, 54 patients were enrolled, and the PFS at 6 months was 57% (31/54, 90% CI 45–68%). The objective response rate (ORR) was 48% (95% CI 35–61%). Additionally, 36 participants experienced grade 3 or higher immune-related adverse events. This included neutropenia 13%, nausea 11%, anemia 7%, and fatigue 6% [18]. Following these results, the pivotal phase III trial was designed, the DREAM3R trial, which compares chemotherapy with durvalumab versus chemotherapy alone (Table 2) [3,17,22,30]. The DREAM3R trial is currently active, with results highly anticipated. 

Moreover, data from the active, open-label, randomized phase III BEAT-Meso study, which contrasts the effectiveness of Atezolizumab, and bevacizumab paired with carboplatin-pemetrexed vs. bevacizumab mixed with carboplatin-pemetrexed, is forthcoming [17]. Other ongoing trials (Table 2) are phase II/III by the Canadian Cancer Trials Group [31], which aims to evaluate pembrolizumab either alone or combined with first-line chemotherapy vs. chemotherapy alone, and a dedicated trial in the Japanese population KEYNOTE-17 is running with a primary endpoint to assess the drug limiting toxicity effects of Pembrolizumab when used with chemotherapy [32]. 

**Table 2 cancers-15-02940-t002:** Ongoing trials for immunotherapy in the frontline.

Trial	Year	Phase	Intervention	Size	Endpoint
KEYNOTE-A17	2022	IB	Pembrolizumab plus chemotherapy ^1^	19	Drug limiting toxicity/AE ^4^
NCT05324436	2022	Cohort	Nivolumab plus Ipilimumab	50	Safety/AEs ^4^
DREAM3R	2021	III	Durvalumab plus chemotherapy ^1^	480	OS ^3^
BEAT-Meso	2019	III	Atezolizumab, Bevacizumab plus chemotherapy ^2^	401	OS ^3^
NCT02784171	2016	II/III	Pembrolizumab plus chemotherapy ^1^	520	PFS ^5^ and OS ^3^
Checkmate743	2016	III	Nivolumab plus Ipilimumab	605	OS ^3^
PrE0505	2016	II	Durvalumab plus chemotherapy ^1^	55	OS ^3^

^1.^ Chemotherapy: Cisplatin plus Pemetrexed. ^2.^ Chemotherapy: Carboplatin plus Pemetrexed. ^3.^ Overall survival. ^4.^ Adverse events. ^5.^ Progression-free survival. References: KEYNOTE-A17 [32], NCT05324436 [33], DREAM3R [30], Beat-Meso [34], NCT02784171 [31], Checkmate743 [27], PrE0505 [35].

## 4. Alternative Checkpoint Inhibitors and Novel Combinations

Though PD-1/PDL-1 and CTLA-4 checkpoint blockades have dominated the clinical development of immune therapy for cancer, there has long been the appreciation that these are not the only proteins that function as immune checkpoints in human biology. As such, there are several other approaches that are being investigated in preclinical and early clinical studies of mesothelioma, given a considerably high number of genetic alterations have been detected. Such alterations may lead to producing neoantigens and the clonal expansion of Tumor-infiltrating T lymphocytes, supporting the concept of MPM as an immunogenic disease [17]. According to genomic studies, the most frequent molecular occurrence in MPM is the loss of tumor suppressor genes. Breast cancer-associated protein 1 gene (BAP1), neurofibromin 2 gene (NF2), cyclin-dependent kinase inhibitor 2A gene (CDKN2A), and occasionally tumor protein p53 gene are examples of frequently inactivated tumor suppressor genes [2,28,36,37]. This premise has encouraged the development of multiple trials that would facilitate the understanding of MPM molecular aberrations and identify those biomarkers that could eventually have diagnostic usefulness and serve as predictors of therapy response through immunohistochemical analysis as well as be targeted through therapy [36,38]. 

The tumor microenvironment enriched with cytotoxic T lymphocytes has also opened the door for the search for biomarkers with their associated targets. Such examples are VISTA (also known as VSIR or B7H5), an immune checkpoint protein, and LAG-3, a T cell receptor that suppresses both T cell activation and cytokine secretion, making it an exciting target for immunotherapy research (Figure 1). Interestingly, even though mesothelial cells do not have LAG-3 receptors, tumor-infiltrating lymphocytes in pleural effusions from patients with mesothelioma demonstrate high amounts of LAG-3 [2,39]. There is a phase I trial, the first in a human open-label study for the MGD013 molecule that is designed to bind LAG-3 and PD-1 in order to maintain or restore the function of T cells [40]. Furthermore, TIM-3 (T cell immunoglobulin and Mucin domains containing protein 3) and OX40/OX40L (members of the TNF receptor superfamily are key co-stimulators of T cells during infection) are currently under investigation (Table 3) [2,39,41]. 

Another ongoing trial is the open-label, non-randomized, phase II single-arm trial where patients will receive Bintrafusp alfa (M7824), a bifunctional fusion protein composed of the extracellular domain of the transforming growth factor B fused to a human immunoglobulin G1 antibody blocking PDL-1 [42].

### 4.1. Other Immune Checkpoint Protein

B7-H3 (CD276) is another immune-checkpoint protein that has been implicated in immune escape. It is ubiquitously expressed in a variety of tumor types as well as MPM. We and others have demonstrated through immunohistochemical testing how B7-H3 is overexpressed in MPM cells [52]. As is the case with mesothelin, several approaches to targeting this protein for advanced solid tumors, including mesothelioma, are being explored; however, in contrast to mesothelin, it is also highly expressed in sarcomatoid tumors and represents a promising target for future therapy. CAR-T and CAR-NK (chimeric antigen receptor-natural killer) therapy targeted to B7-H3 (Figure 1) is already underway for patients with advanced solid tumors, though very little data has been presented or published. Likewise, several other pharmaceutical companies are pursuing the development of a B7-H3 targeted T cell engager therapy. MGD009, a humanized molecule that recognizes B7-H3, entered a phase I trial several years ago; however, due to liver toxicity, the trial was halted [53]. It is not clear what the path for this drug might be in the future. Enoblituzumab is a monoclonal antibody directed against B7-H3, whose Fc portion can mediate antibody-dependent cytotoxicity. The results of a phase I trial of enoblituzumab in combination with pembrolizumab were presented at the 2018 Annual Meeting of the Society of Immunotherapy for Cancer (SITC) [54,55]. As this trial included several different tumor types, there was no data presented in patients with MPM; however, there was a slightly higher response than expected to pembrolizumab alone. In patients who had progressed on prior PD-1 therapy, the objective response rate was less than 10%. While these results are not particularly exciting, this does represent a therapy that could be used as a part of an innate immune strategy with NK cells, for example, due to its ability to activate ADCC. Given that B7H3 is highly expressed in MPM, it would be important to ensure that MPM patients are included in future studies of B7H3 immune therapy. 

### 4.2. Antiangiogenic Therapies and Checkpoint Inhibitors

The introduction of antiangiogenic drugs alongside standard therapy was noted to be a promising development, with the MAPS phase III trial reporting an improvement in overall survival with the addition of bevacizumab compared to chemotherapy alone [56]. Though statistically significant, the survival benefit was quite modest, and while it gained some use in the United States, it was not FDA-approved; however, the role of anti-VEGF therapy for MPM was established. More recently, the LUME-Meso trial was a phase 2 study of the Anti-VEGFR antibody nintedanib, which also demonstrated an improvement in progression-free survival of 9.4 months compared to 5.7 with placebo plus chemotherapy; however, a phase III trial where 541 patients were evaluated did not meet its primary endpoint [17,57].

With the high expression of VEGFR2 observed in mesothelioma cells, tyrosine kinase inhibitor therapies continue to be an attractive option for further research, especially when combined with PD-1/PD-L1 blocking agents. There have been numerous clinical data showing the benefit of a combination of VEGFR-targeted therapy with ICI that has led to developing research in mesothelioma [58,59]. The PEMMELA study was a phase 2 study evaluating the combination of multi-targeted VEGFR2 tyrosine kinase inhibitor, Lenvatinib with pembrolizumab in patients with MPM in the 2nd line after the failure of first-line chemotherapy [43]. Preliminary results were presented at WCLC 2022, showing an ORR of 58%, meeting its primary endpoint. Though there were no unexpected toxicities, most patients required a dose reduction of Lenvatinib due to toxicity (76%). More recently, a trial combining ramucirumab, an anti-VEGFR2 antibody, with nivolumab has been completed, and the results of this study are anticipated soon [50].

## 5. Mesothelin-Targeted Therapy 

Mesothelin is a 40-kilodalton (KD) glycoprotein that is expressed at a low level on mesothelial tissues but is highly expressed on many solid tumors, including MPM. It has been felt to be an attractive target for therapy as it has been implicated in mediating tumorigenesis and disease progression in multiple preclinical models. There have been many strategies for targeting mesothelin in clinical trials. These have included antibodies, radioimmunoconjugates, and antibody-drug conjugates, but for the purpose of this manuscript, we will focus on the immune therapy approaches that have utilized mesothelin as a target.

### 5.1. Immunotoxin Therapy

For many years, the National Cancer Institute (NCI) has been working on a mesothelin-targeted immunotoxin. SS1 anti-mesothelin antibody conjugated to pseudomonas exotoxin was initially used in clinical trials. Though there was some success with this, it was immunogenic due to the murine origin of SS1. LMB-100 is a fully humanized anti-mesothelin antibody conjugated to Pseudomonas exotoxin, which is now in clinical testing [60]. The phase I trial was completed with stable disease as the best response to therapy in 50% of the patients treated. There were no objective responses; however, in 11 patients who subsequently were treated with anti-PD-1 therapy, four had a prolonged disease response, suggesting that the LMB-100 may have stimulated the immune response. As a result, a combination trial of LMB-100 with pembrolizumab is currently ongoing [61]. 

### 5.2. Vaccine Therapy

An anti-mesothelin vaccination strategy has been employed as well. CRS-207 is a live-attenuated Listeria monocytogenes engineered to produce a mesothelin vaccine that showed promising data preclinically [60]. In clinical trials, CRS-207 has been used in combination with chemotherapy and with pembrolizumab. In the phase I study of CRS-207 monotherapy, no objective tumor response was seen, but there was evidence of activation of mesothelin-specific T cells [62]. A subsequent clinical trial of CRS-207 in combination with pembrolizumab revealed no objective response in the first ten patients treated, resulting in the study’s termination [63]. CRS-207 was also combined with platinum/pemetrexed chemotherapy for patients with MPM in a phase 2 trial. Of the 35 patients enrolled, 20 experienced an objective response (57%), though this was a single-arm trial utilizing highly effective chemotherapy. Even though definitive conclusions regarding the importance of CRS-207 cannot be drawn, evaluation of the tumor post-treatment did show some increase in T cell infiltration, suggesting that CRS-207 might have some immune benefits in combination with chemotherapy [64]. CRS-207 is currently undergoing trials in combination with nivolumab and ipilimumab in patients with mesothelin-expressing pancreatic cancer NCT03190265 [65].

### 5.3. Chimeric Antigen Receptor Therapy

Since 2011, CAR-T (chimeric antigen receptor-T cells) studies targeting mesothelin have been ongoing, and these trials have evolved over time. These studies have all been phase I trials using slightly different constructs of anti-mesothelin CARs. Studies have been carried out with and without lymphodepleting chemotherapy and have included other tumor types that express mesothelin, such as NSCLC, pancreatic and ovarian cancers as well as MPM. Furthermore, while initial studies were conducted with CAR-T monotherapy, more recent studies have been conducted in combination with immune checkpoint blockade. 

Initial trials were plagued by a lack of persistence of CAR-T cells after infusion [66,67]. This was felt, in part, to be possibly immunologic due to the use of a murine-derived mesothelin, SS1. This led to the development of fully human mesothelin CARs that are now being tested in clinical trials. Though most of these efforts have not been published, a manuscript published very recently describes the experience with regional delivery of mesothelin-targeted CAR-T [68]. This construct also incorporated an inducible caspase as a suicide gene in the case of excess toxicity.

In this phase I trial, regional CAR T cell implementation was used as MPM is generally a locally aggressive disease and typically does not disseminate systemically. Furthermore, preclinical data from an orthotopic murine MPM model showed that regional administration had superior efficacy at a lower dose by avoiding the sequestration of CART cells in the lungs and by increasing CD4 T cell function. It was also discovered in preclinical studies that PD-1 expression may contribute to CAR-T therapy resistance, which could be overcome by anti-PD1 therapy. Thus, this trial also incorporated systemic administration of pembrolizumab, which was initiated 4 weeks after the infusion of CAR-T cells. MPM patients were enrolled and treated with infusions of cells. Remarkably, there was very little severe toxicity from this regimen. There was no grade 5 toxicity. Cytokine release syndrome (CRS) was seen in 7 patients, with only 2 patients experiencing grade 2 CRS. Most of the toxicity was related to lymphodepleting chemotherapy. The objective response rate was 12.5%, with the vast majority experiencing stable disease. In the patients receiving pembrolizumab, the median OS was 23.9 months. Notably, CAR-T cells were detectable in the blood >100 days after infusion in 39% of the patients treated. This was increased with the augmented dose of the infusion, and T cell persistence required lymphodepleting chemotherapy [68,69,70]. 

### 5.4. Targeted T Cell Receptor Fusion Construct

Gavocel (TC-210) is a mesothelin-targeted T cell receptor fusion construct (TruC) T cell. Recently, initial results from a phase I trial in mesothelin-expressing tumors became available, and these preliminary findings were presented at the 2021 Annual Meeting of the European Society of Medical Oncology (ESMO). MPM patients were treated with Gavocel following lymphodepleting chemotherapy. Two patients had an objective response, and most patients had some degree of tumor regression. The treatment was mostly safe; however, there were two grade 3 adverse events. Updated analysis has been shared via press release and shows a 21% ORR in MPM patients after the treatment of 22 patients. Interestingly, 21/22 patients had tumor regression [71]. Gavocel will now proceed with phase 2 testing alone and in combination with ICI with the recommended phase 2 dose of 1 × 10^8^ cells/m^2^. Given preclinical data suggesting that PDL-1 expression may lead to exhaustion of the infused cells, the next generation TruC T cell is now entering a phase 1 trial. The same mesothelin-targeted engineered T cell receptor is used in TC-510, but it is also engineered with a PD-1/CD28 switch so that PDL-1 expression on the tumor (either baseline or induced by inflammatory cytokines) results in increased activity of the engineered cells rather than inhibition.

Though these are exciting results, the data still needs to be confirmed in subsequent studies. Table 4 summarizes ongoing efforts at developing mesothelin-targeted adoptive cell therapy. It is worth noting that, despite being overexpressed in the majority of MPM cases, mesothelin is almost exclusively found in epithelioid histology. Sarcomatoid histology generally is not amenable to mesothelin-directed therapy; however, there is some evidence that mesothelin-directed therapy may be effective in biphasic mesothelioma, as some of the responders in CAR-T therapy trials had biphasic mesothelioma. This suggests that there is a threshold level of mesothelin expression that is required that has yet to be defined.

## 6. Other Novel Immunotherapy Approaches

### 6.1. NK Cell

Targeted therapy aimed at activating the antitumor activity of Natural Killer (NK) cells is another strategy that could be applied in the future. To date, there are no trials directly using an NK strategy for MPM, but we and others have demonstrated that there is at least some potential for NK cells in the treatment of MPM. Several lines of evidence have suggested that NK cells may play a role in the control of MPM in the tumor microenvironment. First, asbestos has been shown to play a role in decreasing the cytolytic activity of NK cells, suggesting that a lack of active NK cells may play a role in mesothelioma tumorigenesis [79]. Moreover, it has been demonstrated that the cytolytic activity of NK cells can be revived using cytokines such as IL-15 or IL-2 [80]. There is some preclinical data suggesting IL-15 may be useful in combination with an anti-PDL-1 antibody to activate NK cells against mesothelioma targets [81]. This is a strategy that could be moved into clinical testing for MPM. Similar to T cell engagers (Figure 2A), NK cell engagers are in development in the preclinical space (Figure 2B), using either mesothelin or B7-H3 as the tumor target [52]. The first phase I trial of the B7-H3 NK cell engager is expected to be initiated in 2024, which will include several solid tumors. 

### 6.2. Dendritic Cells

Research is currently investigating the use of intra-tumoral injections and dendritic cell (DC) therapy alone or in combination with immunotherapy. The DENIM trial, a phase II/III multicenter randomized study using dendritic cells loaded with allogeneic tumor cell lysate, intends to show how dendritic cell therapy may represent a novel therapeutic approach with promising results in mouse models. This study wishes to examine the efficacy of DC therapy in MPM patients who received first-line treatment with chemotherapy [39,83]. MESOVAX is another interesting trial investigating Pembrolizumab plus autologous dendritic cells in patients with PD-L1-negative advanced mesothelioma who have failed prior therapies. This is an exploratory, single-arm, open-label, phase 1b clinical trial NCT03546426 [84]. 

### 6.3. Oncolytic Virus Therapy

Oncolytic viruses have been known to preferentially replicate and kill cancer cells for many decades. It is only more recently that their potential as an immunotherapy has been appreciated. Since most patients have pre-existing neutralizing antibodies to many commonly used oncolytic viruses, systemic administration of these agents has been limited for most solid tumors. MPM, however, has been considered a prime target for oncolytic virotherapy, given that it is more commonly locally aggressive rather than systemically metastatic. Thus, several viruses have been studied in preclinical settings of MPM [85,86,87]. Furthermore, advances in genetic engineering have enabled the creation of recombinant viruses with immune-activating transgenes. Several reviews have outlined the use of oncolytic viruses in MPM [88]. Though responses have been limited, there have been a few examples of durable responses in patients with MPM [89]. To date, there is minimal data utilizing combination approaches of viruses with ICI or other immune therapies, but given the mechanism of oncolytic viruses in stimulating an immune response, this could be a path toward future development in MPM. Interesting data have been published that also show the potential of oncolytic virotherapy to enhance the infiltration of mesothelin-targeted T cells in the immune microenvironment [90,91]. Thus, there may be a future role of utilizing oncolytic viruses in combination with adoptive cell therapy to make the tumor microenvironment conducive to cell therapy.

## 7. Discussion

Significant advances in the treatment of malignant pleural mesothelioma have resulted from the development of molecular diagnostic testing and an improved understanding of tumor biology. Multiple trials have demonstrated that checkpoint inhibitors have the potential to create a paradigm shift in the treatment of mesothelioma, with clear gains in overall survival, as demonstrated by the CHECKMATE743 trial. This opened the door for new studies investigating other CTLA-4, PD-1/PDL-1 blockers with highly anticipated results, such as those from DREAM3R looking into durvalumab combined with chemotherapy. 

Clearly, tumor histology appears to play an important role in therapy response. The sarcomatoid histology type may respond best to PD-1 antibody treatment; however, the number of non-epithelioid histology patients enrolled in these trials is insufficient to reach a consensus. Studies have also demonstrated that mesothelin-targeted therapies may be more effective in epithelioid histology than in sarcomatoid, with some evidence also suggesting that there could be efficacy in the biphasic subtype. In contrast, B7-H3, a novel immune checkpoint protein, is highly expressed in sarcomatoid histology; if ongoing trials are effective in terms of improving overall survival, this could become the optimal treatment for this type of mesothelioma. Moreover, B7-H3 appears to be one of the most promising checkpoint proteins due to its overexpression in MPM cells along with the advances that have been made with other solid tumors; it is of utmost importance to ensure that MPM patients are included in trials investigating B7-H3 immunotherapy in the years to come. 

Even though antibodies against PD-1/PDL-1 dominate the field, it continues to be intriguing how PDL-1 expression has not proven to have a bearing on treatment response or prognosis, and further clarification, is required. The identification of predictive biomarkers of ICI effects is an unmet need. Compared to other cancers, mesothelioma biomarker research has made limited progress. Some single-arm ICI investigations demonstrate a correlation between response and PD-L1 expression. However, because insufficient survival data was generated, more studies are required to confirm the predictive value of PD-L1 immunohistochemistry for the OS effect.

The hypothesis that mesothelioma is an immunogenic disease with frequent molecular occurrences and a microenvironment rich in cytotoxic T lymphocytes has paved the way for the study of other classes of novel therapies targeting different T cell biomarkers. Despite the failure of antiangiogenic therapies at improving overall survival in MPM, the high expression of VEGFR2 on mesothelioma cells helps the tyrosine kinase inhibitor therapies to remain an attractive option for future research, particularly when combined with PD-1/PDL-blocking agents as is occurring in new ongoing clinical trials. 

Furthermore, therapies aimed at activating the endogenous antitumor function of natural killer cells, utilizing dendritic cells or oncolytic viruses, are under active research, and various strategies have now been developed that may open a path for their implementation in the future. Several lines of evidence suggest that NK cells may play a role in the tumor microenvironment, with asbestos inhibiting the cytolytic activity of NK cells and some data indicating that IL-15 in combination with an anti-PDL-1 antibody may be effective in re-activating NK cells against mesothelioma.

Moreover, mesothelin is a promising target for therapy due to its role in tumorigenesis and disease progression in preclinical models. Early phase trials looking into immunotoxins, vaccine therapy, chimeric antigen receptor therapy, and targeted T cell receptor fusion construct are currently being implemented, with some advancing into phase II trials. 

## 8. Conclusions

Although significant progress has been made on several fronts, mesothelioma research continues to face a myriad of obstacles. Even though multiple trials are underway, the greatest obstacles continue to be their small sample sizes and ongoing selection bias leading to remaining uncertainty over the outcomes of various treatments. There are also obvious impediments in the implementation of novel treatments and in fomenting high-quality research in underdeveloped healthcare systems where the burden of this disease remains unacceptably high. Much work remains to address these inequities and achieve full representation in scientific studies. Nevertheless, this current field holds promise for potential breakthroughs which could alter the course of this disease for the foreseeable future.

## Figures and Tables

**Figure 1 cancers-15-02940-f001:**
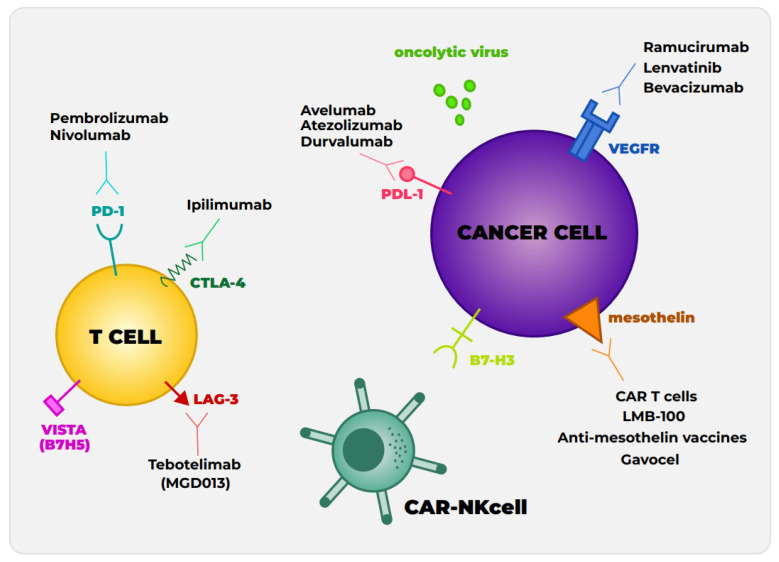
T cell illustrated with its pertinent receptors PD-1, CTLA-4 with their directed immunotherapies, and other innovative biomarker receptors LAG-3 and VISTA currently been researched. Cancer cell with its investigated receptors and their therapies have been studied in preclinical and clinical studies. CAR-NK (chimeric antigen receptor-NK) cell targeting B7H3.

**Figure 2 cancers-15-02940-f002:**
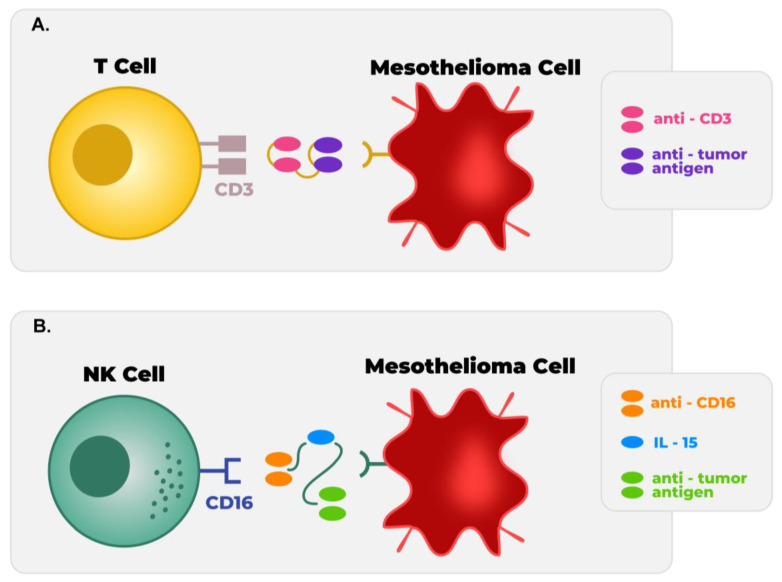
*BiKEs and TriKEs.* (**A**) T cell BiKE (Bispecific Killer cell Engager) immunomodulator with 2 antibody fragments, a recognizing tumor antigen and another fragment targeting CD3 receptor, together, they trigger an antibody and cell-dependent mediated toxicity. (**B**) Natural Killer cells TriKEs (Trispecific Killer cell Engagers) with an antibody fragment recognizing a tumor antigen and a second antibody fragment directed against CD16 on NK cells. IL-15 has been integrated into this model to drive NK cell expansion and enhance NK cell-mediated tumor rejection [82]. Undergoing studies evaluating the potential of these combinational therapies with checkpoint inhibitors and BiKEs and TriKEs against tumor-specific antigens are undertaking and should be explored in MPM.

**Table 3 cancers-15-02940-t003:** Ongoing trials for alternative checkpoint inhibitors and novel combinations.

Trial	Year	Phase	Intervention	Size	Endpoint
NCT05005429	2021	II	Bintrafusp alfa	47	PFS ^4^
PEMMELA	2020	II	Pembrolizumab and Lenvatinib	36	ORR ^5^
NIPU	2020	II	Nivolumab plus Ipilimumab with UV1 vaccine	118	PFS ^8^
NCT04013334	2019	II	MTG201 ^1^ plus Nivolumab	12	ORR ^5^
NCT04040231	2019	I	Galinpepimut-S ^1^ plus Nivolumab	10	MTD ^6^
MESO-PRIME	2019	I	Pembrolizumab plus HSR ^2^	18	DLT ^7^
NCT03894618	2019	I	SL-279252 (PD1-Fc-OX40L) ^3^	87	MTD ^6^/Safety
NCT03502746	2018	II	Nivolumab plus Ramucirumab	35	ORR ^5^
NCT03074513	2017	II	Atezolizumab plus Bevacizumab	137	ORR ^5^
NCT03219268	2017	I	MGD013	353	MTD ^6^AEs ^8^

^1.^ Intra-tumoral Injection. ^2.^ Hypofractionated Stereotactic Radiotherapy. ^3.^ Agonist redirected checkpoint fusion protein. ^4.^ Progression-free survival. ^5.^ Objective response rate. ^6.^ Maximum tolerated dose. ^7.^ Dose-limiting toxicity. ^8.^ Adverse events. References: NCT05005429 [42]. PEMMELA [43,44], NIPU [45,46], NCT04013334 [47], NCT04040231 [48], MESO-PRIME [49], NCT03894618 [41], NCT03502746 [50], NCT03074513 [51], NCT03219268 [40].

**Table 4 cancers-15-02940-t004:** Ongoing trials for mesothelin targeted therapies.

Trial	Year	Phase	Intervention	Size	Endpoint
NCT05451849	2022	I/II	Mesothelin TruC T cells	115	RP2D ^1^ (Phase I)/ORR ^4^ and DCR ^6^ (Phase II)
NCT04840615	2021	I	LMB-100 plus Ipilimumab	20	AEs ^2^/DLT ^3^
NCT04577326	2020	I	Mesothelin CAR T engineered with anti-PD-1	30	MTD ^5^
NCT03907852	2019	I/II	Mesothelin targeted T cells: Gavo-cel	175	DLT ^3^/ORR ^4^
NCT03638206	2018	I/II	Mesothelin targeted T cells	73	AEs ^2^
NCT03054298	2017	I	Mesothelin targeted T cells: huCART-mesocells	27	AEs ^1^/Clinical benefit
NCT03126630	2017	I/II	Pembrolizumab plus anetumab ravtansine ^2^	110	DLT ^3^/ORR ^4^
NCT02414269	2015	I/II	Mesothelin targeted T cells Plus Pembrolizumab	113	AEs ^2^/Clinical benefit

^1.^ Recommended phase 2 dose. ^2.^ AEs: Adverse events. ^3.^ DLT: Dose-limiting toxicity. ^4.^ ORR: Objective response rate. ^5.^ Maximum tolerated dose. ^6.^ DCR: Disease control rate References: NCT054518496 [72], NCT04840615 [73], NCT04577326 [74], NCT03907852 [71], NCT03638206 [75], NCT03054298 [76], NCT03126630 [77] NCT02414269 [78].

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
