# Peer review of "Success of Checkpoint Blockade Paves the Way for Novel Immune Therapy in Malignant Pleural Mesothelioma"

_cancers, 2023, doi:10.3390/cancers15112940_

Round 1

Reviewer 1 Report

1.       The title seems to be very long and contain repeated phrases.

2.       Please edit references since there are separate references at the end of each sentences.

3.       Please also be careful with using “period” at the end of sentences, figure captions,..

4.       Many abbreviations are not defined in their first appearance, please revise the text accordingly.

5.       Some cohort clinical trials are missing like 10.1001/jamanetworkopen.2021.19934.

6.       Abstract: what is missing is the novelty of the current review and what is new in this work that previously published literature did no cover?

7.       A graphical abstract showing the progress of CPI and their combination in MPM could be invaluable.

8.       Line 13: please define MPM.

9.       KEYWORDS: It seems that some keywords are repeated. Please revise.

10.   Further, Tremelimumab, a monoclonal antibody targeting CTLA-4, has not improved patient survival in a randomised, double blind, placebo-controlled phase 2–3 study, therefore, CTLA-4 seems unnecessary in the list, as is NK cell or DC

11.   In simple summary authors addressed the FDA-approve ICI for MPM, however the first three lines of abstract are misleading and highlight the efficacy rather than approval of the agent. Please revise.

12.   Line 38: authors mentioned that “There are currently no curative approaches for MPM” while they have already mentioned the FDA approval of ICB. Please revise.

13.   Line 44-55: please add reference

14.   Line 53: please replace MPM with “previously untreated unresectable MPM”

15.   57: Please omit “.”.

16.   63,64: Please revise the font.

17.   Line 65-70: please add reference.

18.   Several abbreviations are used within the text including ICI, ICB, CPI for immune check point blockade. Please keep consistency within the text.

19.   Line 77: please define OS, also HR and CI in the next line, and line 108: PFS

20.   Line 71: 2.2. Checkpoint inhibitors: I suggest add a table or diagram showing the progress of the CPI in years.

21.   Line 118: 2.3. Antiangiogenic therapies: this section should relocate to section 4 after CPI since some of the references are recent.

22.   Figure 1: some data are missing like MGD013 molecule for LAG-3.

23.   Please define “CAR-NKcell”, I couldn’t find this word in the text.

24.   Please be consistent in using Capital letters.

25.   Please also clarify if by “T cells” authors mean (TIL).

26.   Table 2: I cannot follow the order of ongoing trial in this table. Please reorder them either based on the starting date or the type of intervention.

27.   Trials like MESOVAX combining the CPI with Autologous dendritic cells are not discussed in the text. Please revise.

28.   Line 244: please define NCI

29.   Line 270: please add the number of clinical trial.

30.   Line 309: Two instead of 2

31.   Line 336: 6.1. B7-H3. Please relocate this section to the CPI. This section is distracting in this part.

32.   Line 388: 6.3. Dendritic cells. Please relocate this section under CPI. You can add a subsection as “combination with CPI” and add this part under this subsection. It is confusing since the name of this trial “MESOVAX” is cited in table 2 under Ongoing trials for immunotherapy and the study is further discussed at the end of the paper. Please revise.

33.   395-397: authors discussed MESOVAX in this section. Please add the trial name in this section.

34.   Line 421-4: please revise the font

35.   Concluding remark seems very short and did not cover the conclusive remark, and future perspectives explicitly. Please revise.

Language needs minor polishing.

Author Response

  1. The title seems to be very long and contain repeated phrases.

Answer: title has been shortened and repeated phrases have been deleted

  1. Please edit references since there are separate references at the end of each sentence.

Answer: References updated

  1. Please also be careful with using “period” at the end of sentences, figure captions

Answer: Carefully reviewed periods.

  1. Many abbreviations are not defined in their first appearance, please revise the text accordingly.
  2.  

Answer: Revised and changed

  1. Some cohort clinical trials are missing like 10.1001/jamanetworkopen.2021.19934.

      Answer: This reference is added.  As well, we have added a sentence discussing this data.

  1. Abstract: what is missing is the novelty of the current review and what is new in this work that previously published literature did not cover?

Answer: Our review adds to a small body of work that reviews the use of immunotherapy for mesothelioma.  The novelty of this manuscript is in the review of novel immunotherapy approaches and potential use of alternative immune checkpoints that are currently being tested in clinical trials for other diseases but could be applicable to MPM.  We and others are developing preclinical data to justify future clinical trials for mesothelioma in these areas.

  1. A graphical abstract showing the progress of CPI and their combination in MPM could be invaluable.

      Answer: Added graphical abstract

  1. Line 13: please define MPM.

      Answer: Defined

  1. KEYWORDS: It seems that some keywords are repeated. Please revise.

      Answer:  Keywords have been revised.

  1. Further, Tremelimumab, a monoclonal antibody targeting CTLA-4, has not improved patient survival in a randomised, double blind, placebo-controlled phase 2–3 study, therefore, CTLA-4 seems unnecessary in the list, as is NK cell or DC

      Answer:  This review discusses the lack of efficacy of CTLA-4 antibodies as monotherapy (tremelimumab in the DETERMINE trial) as well as Ipilimumab efficacy in the context of combination with Nivolumab as is currently standard of care.  Thus, I believe it is important to keep CTLA-4 as a keyword. asNK cells and Dendritic cells I believe we should also maintain in the list only because we discuss these approaches as potential future therapy.  

  1. In simple summary authors addressed the FDA-approve ICI for MPM, however the first three lines of abstract are misleading and highlight the efficacy rather than approval of the agent. Please revise.

      Answer: There is nothing misleading about the first 3 lines of the abstract – it is simply stating the facts that nivolumab and ipilimumab have demonstrated an improvement in overall survival in a randomized trial compared to the previous standard of care.  It is in fact, this finding that has led to the FDA-approval.

  1. Line 38: authors mentioned that “There are currently no curative approaches for MPM” while they have already mentioned the FDA approval of ICB. Please revise.

      Answer: We clarified no curative approaches for advanced disease. The fact that this medication was FDA approved, does not imply it will provide definitive cure and it is in fact not being used as curative-intent treatment.

  1. Line 44-55: please add reference

      Answer: Reference was added  

  1. Line 53: please replace MPM with “previously untreated unresectable MPM”

      Answer: First-line treatment implies patients were previously untreated.  In order to clarify, we have replaced first linet treatment of MPM with “previously untreated unresectable MPM” as requested.

  1. 57: Please omit “.”.

      Answer: Omitted

  1. 63,64: Please revise the font.

      Answer: Font revised and changed.

  1. Line 65-70: please add reference.

Answer: Reference was added.

  1. Several abbreviations are used within the text including ICI, ICB, CPI for immune check point blockade. Please keep consistency within the text.

      Answer: All the abbreviations where abbreviated to ICI (immune checkpoint inhibitor )

  1. Line 77: please define OS, also HR and CI in the next line, and line 108: PFS

      Answer: Defined, overall survival (OS), Hazard ratio ( HR) and period free survival ( PFS)

  1. Line 71: 2.2. Checkpoint inhibitors: I suggest adding a table or diagram showing the progress of the CPI in years.

      Answer: Table was added

  1. Line 118: 2.3. Antiangiogenic therapies: this section should relocate to section 4 after CPI since some of the references are recent.

      Answer: Relocated 

  1. Figure 1: some data are missing like MGD013 molecule for LAG-3.

      Answer: Done

  1. Please define “CAR-NKcell”, I couldn’t find this word in the text.

      Answer: Defined

  1. Please be consistent in using Capital letters.

      Answer: Capital letters changed

  1. Please also clarify if by “T cells” authors mean (TIL).

      Answer: T cells are distinct from TIL in that TILs are tumor-infiltrating T cells.  Unless otherwise specified, T cells do not refer to TILs.

  1. Table 2: I cannot follow the order of ongoing trial in this table. Please reorder them either based on the starting date or the type of intervention.

      Answer: Reordered by date from most recent to last.

  1. Trials like MESOVAX combining the CPI with Autologous dendritic cells are not discussed in the text. Please revise.

      Answer: Trial is discussed in the dendritic cells section, and removed from table to avoid confusion

  1. Line 244: please define NCI

      Answer: Defined, National cancer Institute ( NCI)

  1. Line 270: please add the number of clinical trial.

      Answer: Added

  1. Line 309: Two instead of 2

      Answer: Changed

  1. Line 336: 6.1. B7-H3. Please relocate this section to the CPI. This section is distracting in this part.

      Answer: Relocated

  1. Line 388: 6.3. Dendritic cells. Please relocate this section under CPI. You can add a subsection as “combination with CPI” and add this part under this subsection. It is confusing since the name of this trial “MESOVAX” is cited in table 2 under Ongoing trials for immunotherapy and the study is further discussed at the end of the paper. Please revise.

      Answer: I agree, it was confusing to have them on the table of combination therapies. Specially since this table was focusing on checkpoint inhibitors, I removed them from the table to avoid confusion.

  1. 395-397: authors discussed MESOVAX in this section. Please add the trial name in this section.

      Answer:  Added, NCT03546426

  1. Line 421-4: please revise the font

      Answer: Font revised

  1. Concluding remark seems very short and did not cover the conclusive remark, and future perspectives explicitly. Please revise.

Answer: Agree, conclusion revised and extended, we now developed more of a discussion as opposed to a final conclusion given the extend of the review  

Reviewer 2 Report

The review "Immunotherapy for Malignant Mesothelioma – Success of immune Checkpoint Blockade Paves the Way for Novel Immune Therapy Approaches"  review the current state of immune therapy for MPM, explore the knowledge gaps in the field and discuss ongoing novel immunotherapeutic research in early clinical trials.

There are some notable issues.

1. The authors reviewed the current state of immune therapy for MPM, the Immune infiltration in MPM should be reviewed, before summarized immune therapy for MPM;

2. In Early immunotherapy trials for MPM, the authors should summarized the apply of IL-2 in MPM. In addition, should Antiangiogenic therapies be changed to combination therapy or other;

3.The author should introduce the current status and Options of immunotherapy in MPM;

4.In other novel immunotherapy approaches, Novel combinations and cytokine therapy should be summarized 

5. Predictors of response to immunotherapy in MPM should be summarized;

Manuscript contains many typos and would benefit from proofreading by a native speaker, for example, the language does not express the data clearly

Author Response

1.The authors reviewed the current state of immune therapy for MPM, the Immune infiltration in MPM should be reviewed, before summarized immune therapy for MPM;

Answer: I believe a summary of the immune microenvironment in MPM is beyond the scope of this trial.  However, we have introduced the topic and referenced a recent review article highlighting the immune microenvironment in MPM.   

  1. In Early immunotherapy trials for MPM, the authors should summarized the apply of IL-2 in MPM. In addition, should Antiangiogenic therapies be changed to combination therapy or other;

Answer:IL-2 therapy was added. Antiangiogenic therapy was changed to alternative checkpoint inhibitor section.

3.The author should introduce the current status and Options of immunotherapy in MPM;

Answer: On immunotherapy in the frontline. Section 3. It was added how no other immunotherapy besides Nivolumab/ipilimumab has been officially approved.

4.In other novel immunotherapy approaches, Novel combinations and cytokine therapy should be summarized. 

Answer: Table 2 was changed to display novel combinations. Cytokine therapy with Il-15 is  the cartoon, IL-2 and IFN are both discussed in the first section.

  1. Predictors of response to immunotherapy in MPM should be summarized.

Answer: This was  explained more in detail in the discussion

Reviewer 3 Report

This is a well written, comprehensive, well organized review paper discussing the  current  literature and  on going/upcoming clinical trials  in the treatment of malignant mesothelioma,  focusing on immunotherapy  And its combination.

I see you have  data tabulated for upcoming studies.  My only advise would be to add a table with studies that have resulted, showing outcomes, n, type/design.  This will be a good summary in tabular form for enhancing  the quality of your review paper.

Author Response

This is a well written, comprehensive, well organized review paper discussing the  current  literature and  on going/upcoming clinical trials  in the treatment of malignant mesothelioma,  focusing on immunotherapy  And its combination.

I see you have  data tabulated for upcoming studies.  My only advise would be to add a table with studies that have resulted, showing outcomes, n, type/design.  This will be a good summary in tabular form for enhancing  the quality of your review paper.

Answer: Table summarizing most relevant trials leading to present FDA approved treatment mentioned in the text.

Reviewer 4 Report

The manuscript comprehensively summarized current progress on mesothelioma immunotherapy and the ongoing research of the field, with a focus on clinical settings. The manuscript was very informative and well written. Some minor issues were found in Fig. B. The cartoon of IL-15 should be illustrated with one ball (because IL-15 is one single peptide protein). However, some research papers may use fusion of IL-15 and its receptor IL-15Ra or truncated IL-15Ra (named IL-15Ra Sushi) as therapeutics, in this case, two different balls may be used. Still here, Mesothelioma cell (not mesothelin cell).  Anti-CD3, anti-tumor antigen, Anti-CD16. 

Author Response

The manuscript comprehensively summarized current progress on mesothelioma immunotherapy and the ongoing research of the field, with a focus on clinical settings. The manuscript was very informative and well written. Some minor issues were found in Fig. B. The cartoon of IL-15 should be illustrated with one ball (because IL-15 is one single peptide protein). However, some research papers may use fusion of IL-15 and its receptor IL-15Ra or truncated IL-15Ra (named IL-15Ra Sushi) as therapeutics, in this case, two different balls may be used. Still here, Mesothelioma cell (not mesothelin cell).  Anti-CD3, anti-tumor antigen, Anti-CD16. 

Answer: Cartoon was changed as requested

Round 2

Reviewer 1 Report

Dear professor Samuel C. Mok

Editor in chief

Thanks for sending the authors response.

Most of the revision was done and I believe that it can be accepted in its present form, however, minor English language editing is still required to reach the level of satisfaction. 

Regards

Dear professor Samuel C. Mok

Editor in chief

Thanks for sending the authors response.

 Minor English language editing is still required to reach the level of satisfaction. 

Regards